# A Comprehensive Appraisal of Meta-Analyses of Exercise-Based Stroke Rehabilitation with Trial Sequential Analysis

**DOI:** 10.3390/healthcare10101984

**Published:** 2022-10-10

**Authors:** Jia-Qi Li, Yu-Wei Sun, Wing-Sam So, Ananda Sidarta, Patrick Wai-Hang Kwong

**Affiliations:** 1Department of Rehabilitation Medicine, The First Affiliated Hospital of Xi’an Jiaotong University, Xi’an 710061, China; 2Department of Rehabilitation Sciences, Hong Kong Polytechnic University, Hong Kong, China; 3Rehabilitation Research Institute of Singapore, Nanyang Technological University, Singapore 308232, Singapore

**Keywords:** stroke rehabilitation, exercise, meta-analysis, trial sequential analysis, gait speed, balance

## Abstract

Meta-analysis is a common technique used to synthesise the results of multiple studies through the combination of effect size estimates and testing statistics. Numerous meta-analyses have investigated the efficacy of exercise programmes for stroke rehabilitation. However, meta-analyses may also report false-positive results because of insufficient information or random errors. Trial sequential analysis (TSA) is an advanced technique for calculating the required information size (RIS) and more restrictive statistical significance levels for the precise assessment of any specific treatment. This study used TSA to examine whether published meta-analyses in the field of stroke rehabilitation reached the RIS and whether their overall effect sizes were sufficient. A comprehensive search of six electronic databases for articles published before May 2022 was conducted. The intervention methods were divided into four primary groups, namely aerobic or resistance exercise, machine-assisted exercise, task-oriented exercise, and theory-based exercise. The primary outcome measure was gait speed and the secondary outcome measure was balance function. The data were obtained either from the meta-analyses or as raw data from the original cited texts. All data analysis was performed in TSA software. In total, 38 articles with 46 analysable results were included in the TSA. Only 17 results (37.0%) reached the RIS. In conclusion, meta-analysis interpretation is challenging. Clinicians must consider the RIS of meta-analyses before applying the results in real-world situations. TSA can provide accurate evaluations of treatment effects, which is crucial to the development of evidence-based medicine.

## 1. Introduction

Meta-analysis is a statistical technique used to synthesise the results of multiple empirical studies through pooled estimates or significance tests [1]. Since Gene Glass introduced the term in 1976, meta-analysis has been widely used in the evaluation of effect size [2]. A well-conducted meta-analysis of an adequate number of studies can provide a robust estimate of treatment effects and odds ratios [3]. Results from studies with larger homogeneous participants may produce less biased conclusions; thus, such results are regarded as higher-level evidence. An increasing number of meta-analyses have been conducted on the topics of education, psychology, biomedical sciences, and rehabilitation. However, meta-analyses may also report false-positive results because of insufficient information [4] or random error (repeated significance testing) [5]. Hughes suggested that the effect sizes reported in small trials or early-terminated larger trials tend to be overestimated [6]. Some meta-analyses in the field of stroke rehabilitation may potentially overestimate the treatment effect because the number of included studies is insufficient. This is exacerbated by the typically small sample size of clinical rehabilitation trials.

Trial sequential analysis (TSA) evolved from group sequential analysis, which was introduced by Armitage [7] and Pocock [8] in the 1960s, and was further developed by Lan and DeMets [9]. Similarly to the sample size calculation for randomised controlled trials (RCTs), TSA estimates the number of randomised participants required to achieve the statistical power for detecting the desired effect size. If the required sample size has not been reached, TSA provides an adjusted statistical threshold for evaluating intervention effects [10].

One of the key metrics provided by TSA is the *required information size* (RIS), which refers to the number of events or participants required for the detection of a predefined effect size in a meta-analysis [10]. In a random-effects model, high heterogeneity leads to uncertainty, which causes meta-analyses with small samples to prematurely determine statistical significance and thus potentially overestimate the intervention effect. In such cases, the application of the conventional statistical threshold poses a high risk of type I error. Therefore, the threshold for statistical significance must be adjusted on the basis of the accumulated sample size and variability of intervention effects [11]. TSA adopts an *α-spending function* for adjusting the threshold of statistical significance when new information is added to the meta-analysis monitoring boundary, the latter of which refers to a collation of thresholds that this procedure generates [11].

In TSA, a cumulative meta-analysis is performed through the addition of studies in chronological order. As each new study is added to the meta-analysis, the Z-statistic is updated, generating a cumulative Z-curve. The relationship between the Z-curve and the monitoring boundary is assessed to evaluate statistical significance. Here, the Z-statistic is the deviation from an assumed standard normal distribution, where a larger value represents a larger intervention effect. TSA estimates the Z-statistics for studies in chronological order and applies the law of the iterated logarithm to the cumulative meta-analysis of continuous endpoints to indicate the cumulative effect trend. When the last point on the Z-curve lies outside the conventional boundary, the intervention effect is considered to have reached a traditional significance threshold. Firm evidence of an intervention effect can be established if the last point on the Z-curve is outside the monitoring boundary. Otherwise, more trials or greater numbers of participants are required [12].

Although TSA has been used to evaluate the premature reporting of intervention effects in other research areas [13], especially in neonatal medicine [11,14], it has not been systematically used to evaluate meta-analyses in the field of stroke rehabilitation. This study employed TSA to examine whether published meta-analyses on stroke rehabilitation can reach the RIS and whether the overall effect is robust.

## 2. Materials and Methods

### 2.1. Identification and Screening of Studies

Two reviewers (J.-Q.L and W.-S.S) searched the CINAHL, PubMed, Medline, Embase, Scopus, and Cochrane databases. All studies included were published before May 2022 and no restriction was placed on the year of publication. The review protocol was registered in the International Platform of Registered Systematic Review and Meta-analysis Protocols and is available online (registration number: INPLASY202280006). The keywords used were ‘stroke’, ‘exercise training’, ‘meta-analysis’, ‘gait’, and ‘balance’. The details of the search strategies used for each database are presented in Appendix A. One reviewer (J.-Q.L) searched the reference lists of retrieved studies (backward tracking) or searched for articles citing them (forward tracking). Two reviewers (J.-Q.L and Y.-W.S) screened studies for eligibility; the procedure is illustrated in Figure 1. Discussions over eligibility were resolved through discussion with a third reviewer (P.W.K). Walking ability is a standard indicator of impairment in stroke [15] and balance restoration is a major goal of stroke rehabilitation [16]. Therefore, the primary outcome of this analysis was gait performance based either on gait speed measurements or 6-min walk test (6MWT) results. The secondary outcome was balance based on the Berg Balance Scale (BBS).

Meta-analyses were included if they (1) analysed RCTs on stroke and (2) reported gait speed (or 6MWT results) or balance as an outcome. Meta-analyses were excluded if they: (1) were conference abstracts or letters to the editor; (2) did not report statistical parameters such as mean, standard deviation (SD), or number values, and the raw data from the cited studies could not be obtained; or (3) evaluated the effect of exercise training combined with electrical or magnetic stimulation.

### 2.2. Data Extraction

After screening the titles and abstracts, two reviewers (J.-Q.L and Y.-S.W) examined the selected articles and extracted data independently using a standardised data extraction form [17]. The two reviewers (J.-Q.L and Y.-S.W) also extracted the mean, SD, and sample size data from each included study. In the meta-analyses, which investigated both the short-term and long-term effects of treatment, the short-term posttreatment effects were analysed using TSA because of the large sample sizes. For meta-analyses that did not report the data from each study in their forest plots, the raw data were obtained from the original articles and analysed directly in the TSA.

### 2.3. Data Analysis

#### 2.3.1. Statistical Methods

TSA was conducted once for each meta-analysis with a single specific outcome. Gait speed (or 6MWT) and balance were analysed separately in articles reporting both outcomes.

Statistical analyses were performed using TSA software (Trial Sequential Analysis, TSA computer program, version 0.9.5.10 Beta; Copenhagen Trial Unit, Centre for Clinical Intervention Research, Capital Region of Denmark, Copenhagen University Hospital–Rigshospitalet, 2021) [18]. A DerSimonian–Laird random-effects model was used to estimate the effect sizes in the meta-analyses [19]. The data type of a ’continuous’ and ’positive’ outcome was selected as the more positive value indicating a better performance in the outcomes analysed in this study. The individual study in each meta-analysis was added sequentially according to the year of publication. Two boundaries were predefined for evaluating the intervention effects: the (1) conventional and (2) monitoring boundary. Both boundaries were based on a two-sided probability, type I error of 5% and power of 80% (1–β). We used the O’Brien–Fleming-type α-spending function to construct the monitoring boundary [20]. The RIS was calculated on the basis of the effect sizes estimated from the empirical data and adjusted upwards by multiplication by a heterogeneity-adjustment factor [18]. Therefore, large between-trial variations increased the RIS and led to more restrictive monitoring boundaries. The Z-statistics from each trial were used to construct a cumulative Z-curve; we primarily analysed the relationship between the Z-curve and statistical boundaries. Figure 2 illustrates the results of a typical traditional meta-analysis and the TSA.

#### 2.3.2. Outcome Definitions

According to Wetterslev and Jakobsen [14], the significance of meta-analyses can be classified into the following scenarios:
*Potentially spurious evidence of effects*: Analyses whose last point of the Z-curve is outside the conventional boundary (i.e., significant in the original meta-analysis) but inside of the monitoring boundary present potentially spurious evidence, indicating that further trials or larger samples are required. (Figure 3a).*Firm evidence of effects*: Analyses whose Z-curves cross the monitoring boundary but do not reach the RIS present firm evidence of intervention effects (Figure 3b).*Absence of evidence*: Analyses whose Z-curves do not reach the RIS and remain inside the conventional boundary present an absence of evidence. This indicates that, although the result was deemed insignificant, increasing the sample size may lead to a different result (Figure 3c).*Lack of effect:* Analyses whose Z-curves reach the RIS but do not cross the conventional boundary are considered to most likely have no significant effect (Figure 3d).*Verified intervention effects*: Analyses whose Z-curves cross the monitoring boundary and also reach the RIS are considered to have a ‘verified’ intervention effect, indicating that the intervention effects are indeed statistically significant (Figure 3e).

For the first three of the aforementioned scenarios, the additional numbers required to reach the RIS are presented in this review for future studies.

## 3. Results

### 3.1. Eligible Meta-Analyses

We identified 2924 potential studies from the six databases. After the removal of duplicates, 1630 studies remained for title and abstract screening. These articles primarily focused on four types of interventions for stroke rehabilitation: aerobic or resistance exercise; machine-assisted exercise; task-oriented exercise; and theory-based exercise. Theory-based exercise was classified as physical training based on systemic theories such as tai chi, yoga, or Pilates.

Aerobic or resistance exercise included aerobic exercise, high-intensity interval exercise (continuous aerobic training [26]), cardiorespiratory fitness training, exercise programmes (aerobic + resistance), muscle strengthening, and resistance training.

Machine-assisted exercise was any exercise that required an assistive device, such as an exoskeleton, end-effector, or a robot, and was performed on the ground or on a treadmill. Other cases of treadmill training were classified as task-oriented exercise because they can be regarded as a task with steps repeated in a single training session [27]. Exercise programmes that directly involved functional training, including circuit class training, task-oriented exercise, repetitive task training, dual-task cognitive motor training, and other dual-task training were considered task-oriented training [28].

Of the 396 potentially relevant studies, 246 were excluded because they did not perform meta-analyses. Of the remaining 150 studies, 16 studies used aerobic or resistance exercise (Group 1), 16 used machine-assisted exercise (Group 2), 15 used task-oriented exercise (Group 3), and 12 used theory-based exercise (Group 4). Among these 59 studies, the data from only 38 could be extracted for TSA. Studies were excluded from the TSA if they: (1) were conference abstracts [29,30,31]; (2) performed only single-group before–after comparison [32]; (3) lacked raw data, which could not be obtained from the cited articles [33,34,35,36,37,38,39,40]; or (4) did not report the necessary outcome measures (gait speed or balance) [41,42,43,44,45,46,47,48,49].

The monitoring boundary could not be calculated in one study [50] because the actual sample size was much smaller than the estimated RIS and the information was thus insufficient for the TSA software [51,52].

### 3.2. Characteristics of Meta-Analyses

In total, 38 studies were included in the TSA: 28 reported gait speed (including 6MWT) and 18 reported balance function. One study [27] included two comparisons of gait speed in independent and dependent groups, and was analysed twice in the TSA, resulting in two sets of results.

No studies were classified as having a lack of effects. Most of the meta-analyses demonstrated an absence of evidence. Among these were 42.9% of studies reporting gait speed and 33.3% of studies reporting balance outcomes. These were the largest proportion of studies reporting gait speed and the second largest proportion of studies reporting balance outcomes, respectively. Few meta-analyses were classified as having firm evidence, with only 7.1% and 5.6% of such studies reporting gait speed and balance outcomes, respectively. Studies with potentially spurious evidence accounted for 17.9% and 16.7% of those reporting gait speed and balance outcomes, respectively (Table 1).

### 3.3. Meta-Analyses with Potentially Spurious Evidence

Eight studies with significant overall effects were categorised as having potentially spurious evidence in the TSA. Five of these reported gait speed and three reported balance function. The number of additionally required samples ranged from 155 to 421; these studies were only from Group 1 (aerobic or resistance exercise) and Group 3 (task-oriented exercise). Refer to Table 2 and Table 3, where MB is monitoring boundary value.

### 3.4. Meta-Analyses with Firm Evidence of Effects

Three studies with significant overall effects were considered to have firm evidence of their effects in the TSA: two reporting gait speed and one reporting balance function (BBS). All such studies were from Group 3 (task-oriented exercise). Refer to Table 4 and Table 5.

### 3.5. Meta-Analyses with the Absence of Evidence

The results of eleven studies reporting gait speed and six reporting balance function (BBS) were categorised as an absence of evidence in the TSA. This explains the insignificant overall effects reported in the original texts. More information is still needed, especially regarding machine-assisted exercise. Refer to Table 6 and Table 7.

### 3.6. Meta-Analyses with a Lack of Effect

No studies were determined to lack an effect. All of the studies with insignificant Z-curves failed to reach the RIS.

### 3.7. Meta-Analyses with Verified Intervention Effects

Nine studies measuring gait speed and eight measuring balance function reached the RIS and crossed the monitoring boundary in the TSA; thus, their statistical significance and intervention effect were verified. These studies provide evidence supporting the positive effects of rehabilitation (Table 8 and Table 9).

### 3.8. Sensitivity Analyses of the TSA

Studies with a high risk of bias and poor methodology quality could have a negative impact on the validity of the result of a meta-analysis. Sensitivity analyses were conducted by removing studies with a high risk of bias and poor methodology quality. These studies were identified from the risk of bias or quality evaluation in the original meta-analyses. The sensitivity analyses showed that, in general, the required information size decreased, but no difference in the outcome classification was demonstrated for all the included studies.

## 4. Discussion

This is the first large-scale TSA study on stroke rehabilitation. Only 37% of the included meta-analyses achieved the verified effects. One of the possible reasons is the sample size of many included meta-analyses was insufficient due to the nature of the long intervention duration in clinical trials in rehabilitation. Secondly, clinical trials included in the meta-analyses usually demonstrated a large heterogeneity due to the differences in treatment protocols and participants’ characteristics. Thus, this led to a large variation in the effect size estimates.

### 4.1. Discrepancies in the Results of Three Types of Exercise Interventions

Large discrepancies in the analysability were observed among the meta-analyses of the four types of intervention, particularly in the task-oriented exercise group. Among the task-oriented exercise studies—nine results measuring gait speed and seven measuring balance function in this group—only two [61,75] were considered to have verified intervention effects (both reaching the RIS and crossing the monitoring boundaries). In addition, two studies measuring gait speed and one measuring balance function crossed the monitoring boundary before the RIS, indicating a positive treatment effect. The remaining five studies measuring gait speed and six measuring balance require further trials or increased sample sizes to confirm the treatment effect of task-oriented exercise. Although these trials reported statistical significance or insignificance in the original texts, they lacked sufficient statistical evidence according to the TSA results. Thus, concluding whether task-oriented exercises are beneficial for gait speed and balance function in patients with stroke remains challenging. The benefits of this type of exercise require further investigation.

Similarly, the aerobic and resistance exercise studies also exhibited uncertain treatment effects. Of the ten studies reporting gait speed and two reporting balance function, only 25% (three out of twelve) of them achieved verified positive treatment effects. The Z-curves did not cross the monitoring boundaries before reaching the RIS for any of the studies. Additionally, six of the twelve results (five of gait speed and one of balance) were insignificant in the original meta-analyses as well as in the TSA, failing to cross the conventional boundaries (two studies [63,64] had calculated RISs several times the actual sample sizes and insignificant overall effects in the original meta-analyses). Considering a large proportion of insignificant results do not reach the RIS, a potential positive intervention effect of aerobic or resistance exercises on gait speed and balance function cannot be ignored. Stroke rehabilitation RCTs often fail to reach their recruitment targets because of their long intervention durations [82].

Only a small proportion of machine-assisted exercise studies was analysed; only seven studies measuring gait speed and two measuring balance function were included in the TSA. Highly heterogeneous effect sizes were observed in this subgroup, which resulted from the diversity of assistance devices available in clinical practice. Because of the wide variety in participant number and effect size, these meta-analyses had extremely high RISs. The sample sizes of the included meta-analyses differed greatly from the estimated RIS for this subgroup. Future reviews of machine-assisted exercise should focus on more homogeneous machine-assisted exercise interventions or conduct subgroup analyses to obtain more accurate results.

### 4.2. Consistency in the Results of Theory-Based Exercise

Unlike the other types of exercise, theory-based exercise resulted in consistent positive outcomes, except for one study of yoga [70], which included only two trials in its meta-analysis and whose final Z-score did not cross the conventional boundary in the TSA.

One result for gait speed and six for balance qualified as verified treatment effects. Six studies [25,38,79,80,81] investigated the effect of tai chi intervention. However, for the meta-analyses of tai chi, we noted that the included studies were inconsistent among the individual meta-analyses. This is perhaps related to the variations in the inclusion criteria or databases employed. A more standard search strategy is needed for the meta-analysis of the tai chi intervention. Moreover, the number of theory-based exercise interventions implemented for people with stroke is less than those of other exercise treatments. Future studies should develop new theory-based exercises for stroke rehabilitation.

### 4.3. Strengths and Limitations

This study applied TSA to meta-analyses of continuous outcomes. The TSA helped to minimise potential false-positive results by providing a more stringent threshold. This study provides an overview of the robustness of meta-analyses in the field of stroke rehabilitation, which could facilitate clinical decision-making and indicate whether more trials in this area are necessary.

One limitation of this study is that we did not pre-specify the desired effect size for the interventions. A statistically significant result does not necessarily mean the findings are clinically significant or can be directly applied by clinicians.

## 5. Conclusions

In this study, we used TSA to assess whether numerous meta-analyses employed sufficient sample sizes to yield meaningful conclusions. TSA uses a heterogeneity-adjustment factor and provides an RIS to evaluate statistical significance with fewer confounding factors. A total of 38 meta-analyses of exercise interventions for stroke rehabilitation were included in this study. Only a small portion of the results was able to reach the verified treatment effect, indicating that most meta-analyses on exercise-based rehabilitation training required a greater sample size to confirm their results. Future meta-analyses in this area could include a TSA to adjust the statistical threshold or to estimate the required information size to prevent drawing a conclusion prematurely. TSA provides more rigorous evaluations of treatment effects, which are crucial to the development of evidence-based medicine.

## Figures and Tables

**Figure 1 healthcare-10-01984-f001:**
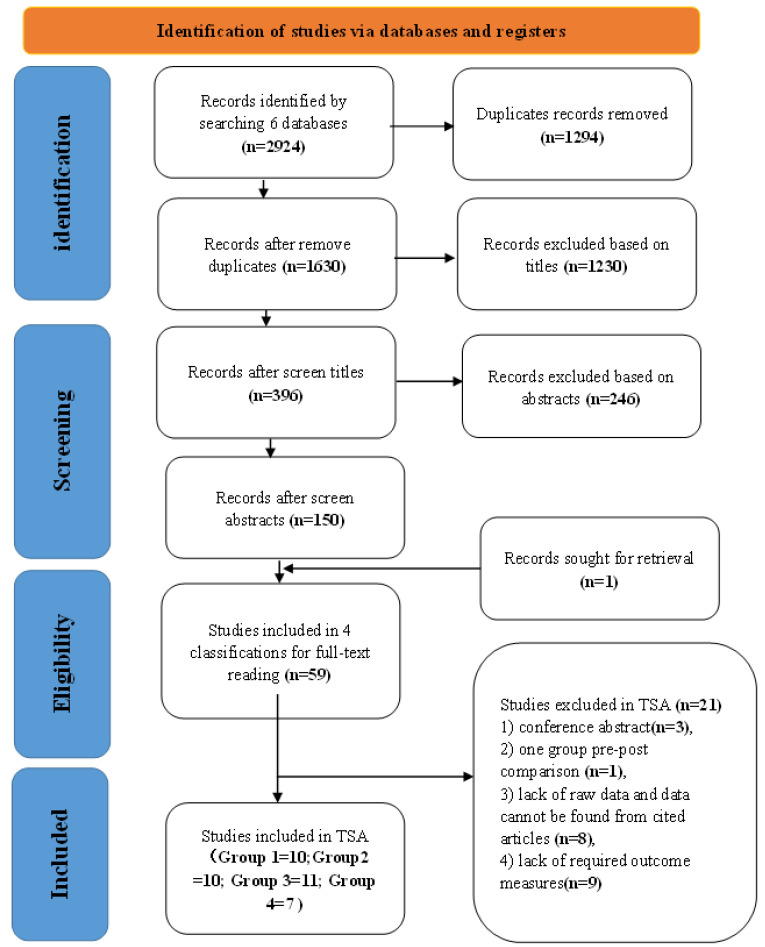
A flow diagram of the study selection.

**Figure 2 healthcare-10-01984-f002:**
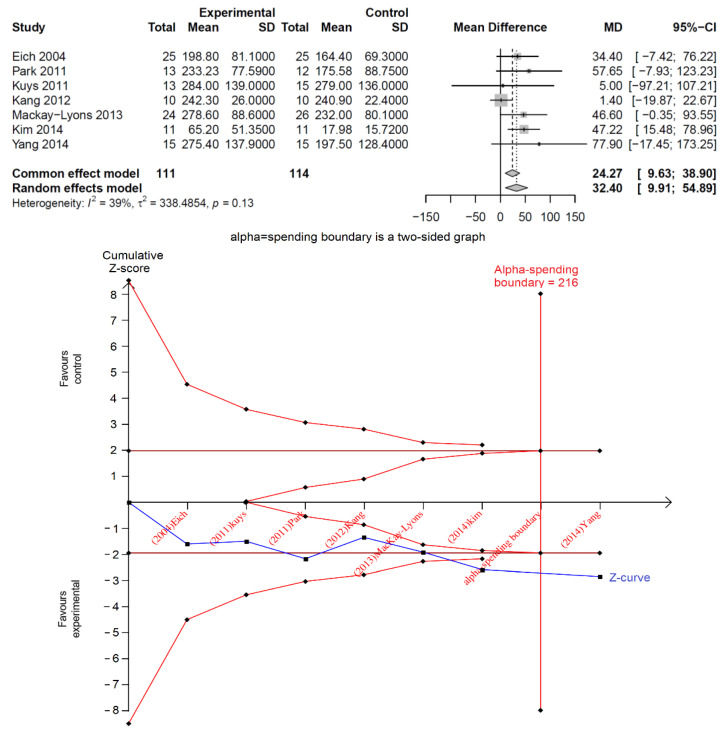
An illustration of a typical forest plot in a meta-analysis in the (**top**) panel and a typical TSA plot shown (**below**) it. Data were extracted from D. H. Saunders, et al. [21].

**Figure 3 healthcare-10-01984-f003:**
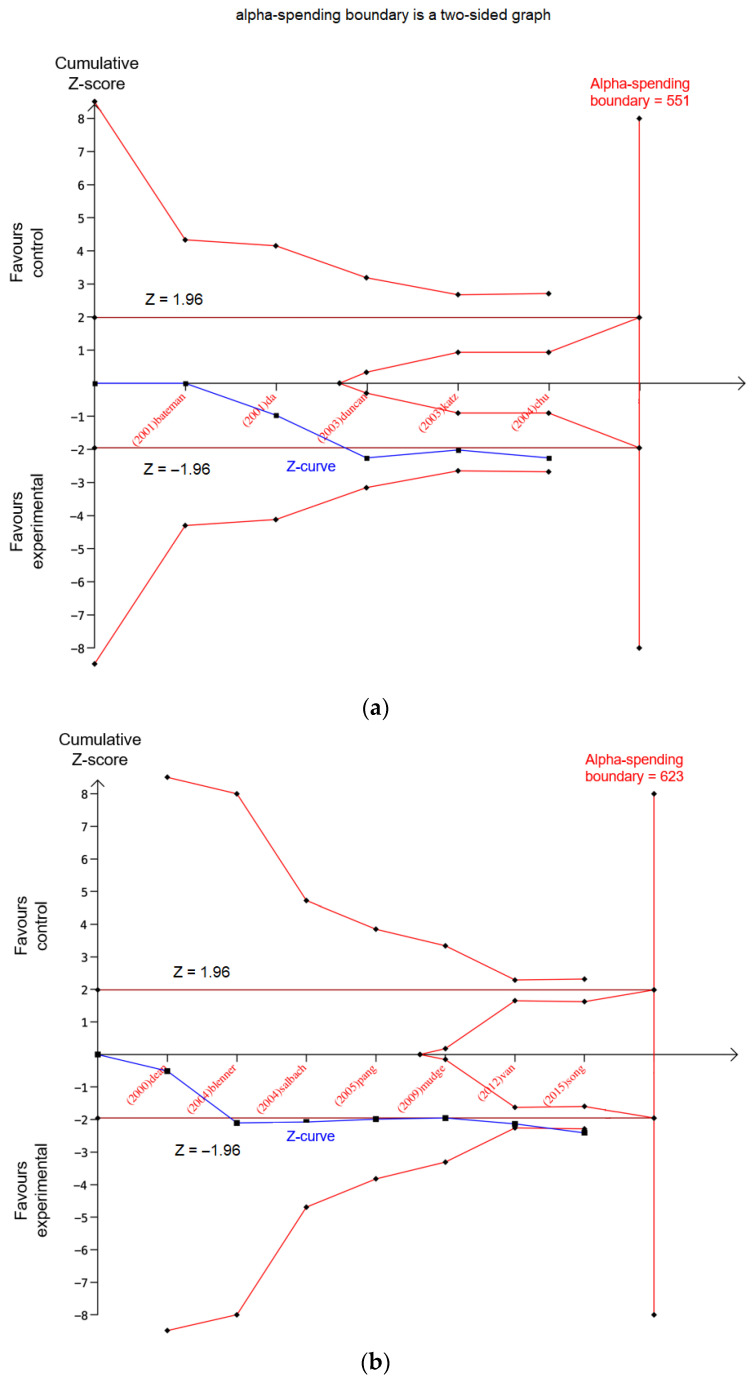
Examples of studies with (**a**) potentially spurious evidence (Pang et al. [22]), (**b**) firm evidence (Bonini-Rocha et al. [23]), (**c**) absence of evidence (Pogrebnoy et al. [24]), (**d**) lack of effects, and (**e**) verified intervention effects (Qin et al. [25]).

**Table 1 healthcare-10-01984-t001:** The TSA results for the included studies.

Situation	Description	Gait Speed	Balance Outcome
Potentially spurious evidence of effects	Cross CB but not MB, and fail to reach RIS	17.9% (n = 5)	16.7% (n = 3)
Firm evidence of effects	Cross CB and MB, but fail to reach RIS	7.1% (n = 2)	5.6% (n = 1)
Absence of evidence	Do not cross CB or MB, and fail to reach RIS	42.9% (n = 12)	33.3% (n = 6)
Lack of effect	Do not cross CB or MB, but reach RIS	0%	0%
Verified intervention effects	Cross CB and MB, and reach RIS	32.1% (n = 9)	44.4% (n = 8)

CB: conventional boundaries; MB: monitoring boundaries; RIS: required information size.

**Table 2 healthcare-10-01984-t002:** Studies with potentially spurious evidence of effects on gait speed.

Group	Review	Interventions	Outcome Measure	No. of Trials (Patients)	Required Information Size	Z-Score (MB)	Additional Information Size Needed
1	M. Y. Pang et al., 2006 [22]	aerobic exercise	gait speed	5 (346)	551	−2.28 (−2.29)	205
1	B. J. Kendall et al., 2015 [53]	aerobic exercise	6MWT	8 (423)	804	−2.03 (−2.89)	381
1	L. Luo et al., 2019 [54]	high-intensity exercise	gait speed	11 (345)	604	−2.02 (−2.82)	259
3	L. Wevers et al., 2011 [55]	circuit class training	gait speed	4 (214)	335	−2.35 (−2.59)	121
3	S. Silva et al., 2020 [56]	task-oriented exercise	gait speed	6 (191)	346	−2.01 (−2.84)	155

**Table 3 healthcare-10-01984-t003:** Studies with potentially spurious evidence of an effect on balance function (BBS).

Group	Review	Interventions	Outcome Measure	No of Trials (Patients)	Required Information Size	Z-Score (MB)	Additional Information Size Needed
3	B. French et al., 2010 [57]	repetitive task training	balance (BBS)	9 (504)	925	−2.05 (−2.89)	421
3	Y. Shu et al., 2022 [58]	dual-task training	balance (BBS)	7 (219)	438	−1.98 (−3.01)	219
3	X. Zhang et al., 2022 [59]	dual-task training	balance (BBS)	6 (194)	369	−2.03 (−2.9)	175

**Table 4 healthcare-10-01984-t004:** Studies with firm evidence of effects on gait speed.

Group	Review	Interventions	Outcome Measure	No. of Trials (Patients)	Required Information Size	Z-Score (MB)	Additional Information Size Needed
3	J. Schröder et al., 2019 [60]	repetitive gait training	gait speed	8 (572)	762	−2.39 (−2.39)	190
3	A. C. Bonini-Rocha et al., 2018 [23]	circuit-based exercises (CBEs)	change in gait speed	7 (516)	623	−2.43 (−2.29)	107

**Table 5 healthcare-10-01984-t005:** Studies with firm evidence of an effect on balance function (BBS).

Group	Review	Interventions	Outcome Measure	No. of Trials (Patients)	Required Information Size	Z-Score (MB)	Additional Information Size Needed
3	Q. Zhou et al., 2021 [61]	cognitive motor dual-task training	change in balance (BBS)	5 (110)	132	−2.56 (−2.23)	22

**Table 6 healthcare-10-01984-t006:** Studies with absence of evidence on gait speed.

Group	Review	Interventions	Outcome Measure	No of Trials (Patients)	Required Information Size	Z-Score (MB)	Additional Information Size Needed
1	I. G. L. Van De Port et al., 2007 [62]	cardiorespiratory fitness training	gait speed	2 (102)	262	−1.74 (−3.44)	160
1	D. Pogrebnoy et al., 2019 [24]	exercise programmes (aerobic + resistance)	gait speed	5 (248)	728	−1.66 (−3.7)	480
1	L. Da Campo et al., 2021 [63]	aerobic exercise (cycle ergometry)	6MWT	3 (188)	1992	0.87 (−8.0)	1804
1	S. Wist et al., 2016 [64]	muscle strengthening	6MWT	6 (265)	1808	−1.07 (−8.0)	1543
1	S. Mehta et al., 2012 [65]	resistance training	6MWT	8 (331)	748	−1.86 (−3.2)	417
2	J. Mehrholz et al., 2017 [27]	machine-assist (dependent at baseline)	6MWT	5 (639)	6016	0.91 (8.0)	5377
2	A. Y. Gelaw et al., 2019 [66]	treadmill assisted gait training	gait speed	8 (695)	7790	0.81 (−8.0)	7095
2	J. Mehrholz et al., 2020 [67]	machine-assist	6MWT	24 (1136)	8346	−0.99 (−8.0)	7300
3	J. C. Polese et al., 2013 [68]	treadmill	6MWT	6 (287)	936	−1.55 (−3.89)	649
3	C. English et al., 2017 [69]	circuit class therapy	gait speed	2 (437)	2567	−1.17 (−5.36)	2130
3	X. Zhang et al., 2022 [59]	dual-task training	gait speed	8 (225)	1328	−1.15 (−8.0)	1103

**Table 7 healthcare-10-01984-t007:** Studies with absence of evidence on balance function (BBS).

Group	Review	Interventions	Outcome Measure	No of Trials (Patients)	Required Information Size	Z-Score (MB)	Additional Information Size Needed
1	L. Da Campo et al., 2021 [63]	aerobic exercise (cycle ergometry)	change in balance (BBS)	3 (195)	513	−1.78 (−3.48)	318
2	A. Y. Gelaw et al., 2019 [66]	treadmill assisted gait training	balance (BBS)	2 (456)	8913	0.64 (−8.0)	8497
3	L. Wevers et al., 2011 [55]	circuit class training	balance (BBS)	2 (154)	1788	0.82 (−8.0)	1634
3	C. English et al., 2017 [69]	task-oriented exercise	balance (BBS)	4 (171)	801	−1.3 (−4.75)	630
3	A. C. Bonini-Rocha et al., 2018 [23]	circuit-based exercises	change in balance (BBS)	3 (174)	2689	0.71 (−8.0)	2515
4	M. Lawrence et al., 2017 [70]	theory-based exercise	balance (BBS)	2 (69)	358	1.23 (5.01)	289

**Table 8 healthcare-10-01984-t008:** Studies with verified intervention effects on gait speed.

Group	Review	Interventions	Outcome Measure	No. of Trials (Patients)	Required Information Size	Z-Score (MB)	Additional Information Size Needed
1	D. H. Saunders et al., 2020 [21]	aerobic exercise	6MWT	7 (225)	216	−2.86 (−1.96)	N/A
1	J. M. Anjos et al., 2022 [71]	high-intensity interval training	change in gait speed	4 (100)	72	−2.87 (−1.96)	N/A
2	L. Ada et al., 2011 [72]	machine-assist	gait speed	4 (258)	117	−4.2 (−1.96)	N/A
2	J. Mehrholz et al., 2017 [27]	machine-assist (independent at baseline)	6MWT	10 (423)	423	−2.81 (−1.96)	N/A
2	M. F. Bruni et al., 2018 [73]	end-effector device	gait speed	7 (469)	441	−2.79 (−1.96)	N/A
2	L. R. Nascimento et al., 2021 [74]	treadmill assist walking	gait speed	6 (266)	88	−4.78 (−1.96)	N/A
3	B. French et al., 2016 [75]	repetitive task training	gait speed	12 (685)	606	−2.93 (−1.96)	N/A
3	Q. Zhou et al., 2021 [61]	cognitive motor dual-task training	change in gait speed	5 (119)	107	−3.02 (−1.96)	N/A
4	Y. Leng et al., 2019 [76]	Pilates exercise	gait speed	2 (80)	65	−3.13 (−1.96)	N/A

**Table 9 healthcare-10-01984-t009:** Studies with verified intervention effects on balance function (BBS).

Group	Review	Interventions	Outcome Measure	No. of Trials (Patients)	Required Information Size	Z-Score (MB)	Additional Information Size Needed
1	J. M. Anjos et al., 2022 [71]	high-intensity interval training	change in balance (BBS)	2 (64)	60	−2.93 (−1.96)	N/A
2	A. Staples et al., 2017 [77]	robotic-assisted gait training	balance (BBS)	4 (108)	69	−3.51 (−1.96)	N/A
4	B.-L. Chen et al., 2015 [78]	traditional Chinese exercise	balance (BBS)	6 (529)	379	−3.61 (−1.96)	N/A
4	L. Qin et al., 2016 [25]	tai chi exercise	balance (BBS)	9 (558)	243	−4.65 (−1.96)	N/A
4	Y. Li et al., 2017 [79]	tai chi exercise	balance (BBS)	9 (670)	152	−6.42 (−1.96)	N/A
4	D. Lyu et al., 2018 [80]	tai chi exercise	balance (BBS)	7 (328)	248	−3.23 (−1.96)	N/A
4	Y. Leng et al., 2019 [76]	Pilates exercise	balance (BBS)	3 (142)	89	−3.59 (−1.96)	N/A
4	X. Zheng et al., 2021 [81]	tai chi exercise	balance (BBS)	5 (376)	196	−3.53 (−1.96)	N/A

## Data Availability

No new data were created or analysed in this study. Data sharing is not applicable to this article.

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
