# Peer review of "A Comprehensive Appraisal of Meta-Analyses of Exercise-Based Stroke Rehabilitation with Trial Sequential Analysis"

_healthcare, 2022, doi:10.3390/healthcare10101984_

Round 1

Reviewer 1 Report

I think it’s a timely and essential systematic review. I would recommend it if the following issues are handled properly.

1. Provide enough details about data analysis in TSA software.

2. Explain why only 37.0% of the analyzed results reached the RIS.

3. Provide one Figure to illustrate the meta-analysis and trial sequential analysis (TSA).

4. Rewrite the conclusion to provide enough significance for this systematic review.

Author Response

Dear Editor and Reviewers,

We have considered each comment carefully and have to the best of our abilities tried to implement the suggested changes in the revised manuscript. The revisions were highlighted in red in the revised manuscript. All the suggestions helped us to evaluate our study even more precisely in order to deliver an improved, high-quality scientific manuscript which we hope will now meet the high standards of 2nd Edition of Neurorehabilitation: Looking Back and Moving Forward. Following this peer review process, we feel the manuscript has improved as such.

Responses to Reviewer 1

  1. Provide enough details about data analysis in TSA software.

 Response: Thank you for the comment. We have supplied the details of conducting the TSA(Page 3). Other prior setting that was used during the TSA have been added in the text. (Page3 Line 124 -126)

  1. Explain why only 37.0% of the analysed results reached the RIS.

 Response: Thank you for the comment. The context has been highlighted in the Discussion section (Page 15). One of the possible reasons is the sample sizes of many included meta-analyses were insufficient due to the nature of long intervention durations in a clinical trial of rehabilitation. Secondly, clinical trials included in the meta-analysis usually demonstrated a large heterogeneity due to the differences in treatment protocol and participants’ characteristics. Thus, this led to a large variation in the effect size estimates. (Page 14 -15, Line 289-295)

  1. Provide one figure to illustrate the meta-analysis and trial sequential analysis (TSA).

  Response: Thank you for the comment. The figure has been added to the Method section (Page 4).

  1. Rewrite the conclusion to provide enough significance for this systematic review.

Response: Thank you for the comment. The context has been added(Page 16). In this study, we used TSA to assess whether numerous meta-analyses employed sufficient sample sizes to yield meaningful conclusions. TSA uses heterogeneity-adjustment factor and provides RIS to evaluate statistical significance with fewer confounder factors. In this study, a total of 38 meta-analyses of exercise interventions for stroke rehabilitation were included. Only a small portion of the results was able to reach the verified treatment effect, indicating that most meta-analyses on exercise-based rehabilitation training required greater sample size to confirm their results. Future meta-analyses on this area could include a TSA to adjust the statistical threshold or to estimate the required information size to prevent drawing a conclusion prematurely. TSA provides more rigorous evaluations of treatment effects, which are crucial to the development of evidence-based medicine. (Page 16 Line 357-367)

Reviewer 2 Report

The manuscript is well though out,  but some parts need to be written in detail.

1.      This study collected 38 articles that used TSA to evaluate the meta-analysis. Are there any potential false positive results?

2.      Why is the bias of individual studies and expected effects of pre-specified interventions not assessed? Can this limitation be further addressed?

3.      Can you elaborate on the reasons why each step study was excluded.

4.      What is the contribution of the research?

5.      What are the future research developments?

Author Response

Dear Editor and Reviewers,

We have considered each comment carefully and have to the best of our abilities tried to implement the suggested changes in the revised manuscript. The revisions were highlighted in red in the revised manuscript. All the suggestions helped us to evaluate our study even more precisely in order to deliver an improved, high-quality scientific manuscript which we hope will now meet the high standards of 2nd Edition of Neurorehabilitation: Looking Back and Moving Forward. Following this peer review process, we feel the manuscript has improved as such.

Responses to Reviewer 2:

  1. This study collected 38 articles that used TSA to evaluate the meta-analysis. Are there any potential false positive results?

 Response: Thank you for the comment. There are 11 results ( with 3 results crossing the monitoring boundaries and 8 results not crossing the monitoring boundaries) that are regarded as false positive results, which means they are significant in the original meta-analyse but fail to reach the required information in the TSA program. (Page 14-15, Line 289-294)

  1. Why is the bias of individual studies and expected effects of pre-specified interventions not assessed? Can this limitation be further addressed?

Response: Thank you for the comment. The content has been added in the result part (Page 12-13). Studies with high risk of bias and poor methodology quality could have a negative impact on the validity of the result of a meta-analysis. Sensitivity analyses were conducted by removing studies with a high risk of bias and poor methodology quality. These studies were identified from the risk of bias or quality evaluation in the original meta-analyses. Results of the sensitivity analyses showed that, in general, the required information size decreased but no difference in the outcome classification was demonstrated for all the included studies. (Page 12 -13, Line 278-284)

  1. Can you elaborate on the reasons why each step study was excluded.

Response: Thank you for the comment. The identification of studies is illustrated in Figure 3 (Page 8). After deleting the unrelated studies according to the titles and abstracts. There are overall 59 studies for full-text reading, 21 studies are excluded in TSA because of 1) conference abstract(n=3), 2) only in one group pre-post comparison (n=1), 3) lack of raw data and data cannot be found from cited articles (n=8), 4) lack of required outcome measures(n=9).

  1. What is the contribution of the research?

Response: Thank you for the comment. The context is added in the conclusion part(Page 16). This research uses TSA to re-analyses the intervention effect of 4 types of exercise in stroke patients. TSA served as a rigorous evaluation tool of treatment effects with additional monitoring boundaries, which subsides the shortage of traditional meta-analyses. The TSA helped to minimise the potential false-positive results by providing a more stringent threshold. The study provides an overview of the robustness of meta-analyses in the field of stroke rehabilitation, which could facilitate clinical decision-making and indicate whether more trials in this area are necessary. (Page 16, Line 348 – 351)

  1. What are the future research developments?

Response: Thank you for the comment. The context is added in the conclusion part(Page 16). The results show that only a small portion of the results reached the verified treatment effect, indicating that most meta-analyses on exercise-based rehabilitation training required greater sample size to confirm their results. Future meta-analyses on this area could include a TSA to adjust the statistical threshold or to estimate the required information size to prevent drawing a conclusion prematurely. TSA provides more rigorous evaluations of treatment effects, which are crucial to the development of evidence-based medicine. (Page 16Line 357 – 367)